



# Assessment of recent advances in measurement techniques for atmospheric carbon dioxide and methane observations

Christoph Zellweger[1], Lukas Emmenegger[1], Mohd Firdaus[2], Juha Hatakka[3], Martin Heimann[4], Elena Kozlova[5], T. Gerard Spain[6], Martin Steinbacher[1], Marcel V. van der Schoot[7], Brigitte Buchmann[8]

[1] Empa, Swiss Federal Laboratories for Materials Science and Technology, Laboratory for Air Pollution / Environmental Technology, CH-8600 Dübendorf, Switzerland.

[2] Atmospheric Science & Cloud Seeding Division, Malaysian Meteorological Department, Ministry of Science, Technology and Innovation, Kuala Lumpur, Malaysia

[3] Finnish Meteorological Institute, Helsinki, Finland

[4] Max Planck Institute for Biogeochemistry, Jena, Germany, and Division of Atmospheric Sciences, Department of Physics, University of Helsinki, Finland

[5] College of Life and Environmental Sciences, University of Exeter, Exeter, UK

[6] National University of Ireland, Galway, Ireland

[7] Oceans & Atmosphere, Commonwealth Scientific and Industrial Research Organisation, Aspendale, Victoria, Australia

[8] Empa, Swiss Federal Laboratories for Materials Science and Technology, Department Mobility, Energy and Environment, CH-8600 Dübendorf, Switzerland.

Correspondence to: C. Zellweger (christoph.zellweger@empa.ch)

**Abstract.** Until recently, atmospheric carbon dioxide ($CO_2$) and methane ($CH_4$) measurements were made almost exclusively using Non-dispersive Infrared (NDIR) absorption and gas chromatography with flame ionization detection (GC/FID) techniques, respectively. Recently, commercially available instruments based on spectroscopic techniques such as Cavity Ring Down Spectroscopy (CRDS), Off-Axis Integrated Cavity Output Spectroscopy (OA-ICOS) and Fourier Transform Infrared (FTIR) spectroscopy have become more widely available and affordable. This resulted in a widespread use of these techniques at many measurement stations. This manuscript is focused on the comparison between a CRDS "travelling instrument" that has been used during performance audits within the Global Atmosphere Watch (GAW) programme of the World Meteorological Organization (WMO) with instruments incorporating other, more traditional techniques for measuring $CO_2$ and $CH_4$ (NDIR and GC/FID). We demonstrate that CRDS instruments and likely other spectroscopic techniques are suitable for WMO GAW stations and allow a smooth continuation of historic $CO_2$ and $CH_4$ time series. Moreover, the analysis of the audit results indicates that the spectroscopic techniques have a number of advantages over the traditional methods which will lead to the improved accuracy of atmospheric $CO_2$ and $CH_4$ measurements.



# 1    Introduction

Long-term observations of atmospheric greenhouse gases (GHGs) such as carbon dioxide ($CO_2$) and methane ($CH_4$) are crucial for the understanding of regional and global GHG budgets and their evolution. This requires data sets traceable to a common reference. Central Calibration Laboratories (CCLs) operate within the Global Atmosphere Watch (GAW) programme of the World Meteorological Organization (WMO) to provide measurement standards on the international calibration scales. The WMO/GAW programme strives to achieve ambitious compatibility goals that enable scientific interpretation of continental or global scale atmospheric observations measured by different laboratories or in-situ stations. The compatibility goals apply to the gas mole fraction ranges observed in the unpolluted troposphere. Currently, these goals stand at ±0.1 ppm for $CO_2$ (±0.05 ppm for the Southern Hemisphere) and ±2 ppb for $CH_4$, whilst the extended goals of ±0.2 ppm and ±5 ppb respectively apply to measurements in more polluted environments (WMO, 2014). Additionally, the traceability of the measurements at GAW stations to the international calibration scales is evaluated by regular system and performance audits by the designated World Calibration Centres (WCCs).

Continuous measurements of $CO_2$ have been available since the 1950s (Harris, 2010; Keeling, 1960), and global methane coverage through direct measurements became available in the late 1970s (Khalil and Rasmussen, 1983; Kirschke et al., 2013). Until only recently, atmospheric $CO_2$ measurements have been made almost exclusively by Non-dispersive Infrared (NDIR) absorption technique (Komhyr et al., 1989) whilst $CH_4$ has been measured by gas chromatography equipped with flame ionization detectors (GC/FID) (Dlugokencky et al., 1995). Over the past few years, spectroscopic techniques such as Direct Absorption Spectroscopy (McManus et al., 2015), Cavity Ring Down Spectroscopy (CRDS) (Chen et al., 2010; Crosson, 2008), cavity enhanced off-axis Integrated Cavity Output Spectroscopy (OA-ICOS) (O'Shea et al., 2013) and Fourier Transform Infrared (FTIR) spectroscopy (Griffith et al., 2012) have become commercially available for the measurements of atmospheric $CO_2$ and $CH_4$. Currently, many traditional NDIR $CO_2$ and GC/FID $CH_4$ systems are being replaced by modern spectroscopic instruments (Brailsford et al., 2012). These new techniques have some clear advantages concerning sensitivity, precision, linearity, time response as well as with regard to the measurement setup. They further require less frequent calibration. However, there exist only a few published studies (Rella et al., 2013; Schibig et al., 2015; Vardag et al., 2014) comparing these modern measurement techniques with $CO_2$ NDIR or $CH_4$ GC/FID, and and crucial information is still lacking to demonstrate that the former can guarantee a smooth continuation of historic and ongoing time series.

Our paper presents a set of comparison experiments of NDIR $CO_2$ and GC/FID $CH_4$ measurements with a CRDS travelling instrument that were made as part of the system and performance audits of the World Calibration Centre for Surface Ozone, Carbon Monoxide, Methane and Carbon Dioxide (WCC-Empa) (Buchmann et al., 2009). In addition, we present $CO_2$ and $CH_4$ comparisons between an OA-ICOS and the travelling instrument. The concept of using the travelling instrument for a comprehensive assessment of atmospheric measurements is encouraged by the recommendations of the WMO/IAEA Meetings on Carbon Dioxide, Other Greenhouse Gases, and Related Measurement Techniques (WMO, 2012, 2014) and has



shown to a highly valuable tool for quality control (Hammer et al., 2013; Zellweger et al., 2013). Our comparison experiments presented here were conducted at four stations within the GAW network which cover different climatological conditions ranging from tropical to subarctic conditions characterised by varied atmospheric water vapour contents. We present and discuss the influence of water vapour on the quality of the atmospheric measurements as most of our

measurement campaigns were conducted without drying of the ambient air samples for the CRDS travelling instrument. Furthermore, we investigate the impact of data coverage on hourly averaged data which represents the standard aggregation period for data submission to most data repositories. Then, we examine the CRDS data with regard to the repeatability of calibration cylinder measurements and discuss calibration strategies. And finally, we analyse the data collected during $CH_4$ and $CO_2$ performance audits over the past few years from the perspective of the measurement techniques used to obtain

them.

## 2   Experimental

The quality assurance strategy of the GAW programme comprises system and performance audits (hereafter only called audit) carried out by World Calibration Centres (WCCs). The concept of the audit procedure has been described in detail elsewhere (Buchmann et al., 2009; Klausen et al., 2003). In brief, WCC-Empa is the designated WCC for $CH_4$ (since 2000)

and $CO_2$ (since 2010) audits. An audit involves the comparison of travelling standards (i.e. compressed gas in high pressure cylinders) on the analytical system of the audited station (WMO, 2011b). The travelling standards are calibrated against primary laboratory reference standards traceable to the Central Calibration Laboratory (CCL) before and after the audit. The audited station's personnel analyse the travelling standards and report the mole fractions which are compared to the values assigned by the WCC. The result is analysed by a linear regression between the reference (WCC) and the station values. For

the calibration of the travelling standards at WCC-Empa, a GC/FID (Varian 3800) system was used from 2000 to 2009 for $CH_4$; from 2009 a CRDS (Picarro Inc., G1301 $CO_2/CH_4/H_2O$ analyser) has been used for both $CH_4$ and $CO_2$ calibrations. Several standards of the CCL (NOAA/ESRL, National Oceanic and Atmospheric Administration / Earth System Research Laboratory) are used as reference standards at WCC-Empa ensuring traceability to the CCL.

Quality assessments based on the comparison of travelling standards alone have their limitations, since they do not cover all

parts of the analytical system that may bias a measurement, such as inlet and drying systems (WMO, 2011a). Furthermore, comparisons of travelling standards during on-site audits as well as round robin experiments are only snapshots potentially biased, e.g. by coincidental instrument malfunction (results worse compared to normal operation) or by extraordinary care taken during analysis (results better compared to normal operation). Therefore, it has been recommended that the quality control procedure during on-site audits should include parallel measurements with a travelling instrument whenever feasible

(WMO, 2011a, 2012, 2014).

The concept of the practical realisation of the on-site data comparison is illustrated in Figure 1. The core part is a CRDS analyser (Picarro Inc., G2401) as travelling instrument (a). An independent inlet system (b) is used during the parallel





measurement with the travelling instrument, and if feasible, the travelling instrument samples from the station and the independent inlet system sequentially (c). Furthermore, the travelling instrument is independently calibrated using its own set of standards (d), which normally comprises a subset of the travelling standards used for the audit. For further confirmation of the compatibility of the two systems, the travelling standards (e) are measured on both the station analyser

and the travelling instrument. Dry air mole fractions are compared for both the travelling standards and the travelling instrument comparisons.

For this study we used the data from two Picarro G2401 $CO/CH_4/CO_2/H_2O$ CRDS instruments (Picarro Inc., USA). The instruments were calibrated every 30-40 hours using dry compressed air as working standard. In most cases the sample air was not dried prior to analysis with the travelling instrument and a humidity correction using the Empa method (Rella et al.,

2013) was applied to all data. The correction function was determined several times for each instrument, and a single function that was initially obtained was used to apply the correction. Yver Kwok et al. (2015) have recently published a comprehensive assessment of the performance of the Picarro G2401 analyser, and our set-up of the travelling instrument was done along the lines of their recommendations. However, in contrast to their approach, we were running calibrations more frequently but only as one cycle. Furthermore, the background signal of both travelling instruments was initially adjusted

using so-called zero air ($CO_2$ and $CH_4$ free natural air). The following calibration strategy was used:

- A working standard (calibrated against certified CCL laboratory standards before and after each campaign) with mole fractions close to ambient air was analysed every 30 – 40 h.
- A LOESS fit was applied to this data.
- The ratio of the assigned working standard value to the LOESS fit was used to apply a drift correction to all data.
- To verify the calibration, two additional cylinders were measured as target standards. The same calibration and water vapour corrections as for ambient air were applied.

The following calibration scales were used: $CO_2$: WMO-X2007 (Zhao and Tans, 2006), $CH_4$: WMO-X2004 (Dlugokencky et al., 2005).

An example of the working and target standard measurements is shown in Figure 2. It can be seen that the variation of the

target gas measurements did not exceed the range of ± half of the WMO compatibility goals of 0.1 ppm for $CO_2$ and 2 ppb for $CH_4$, which is usually required for intra-laboratory repeatability (WMO, 2014). Furthermore, the maximum drift between two consecutive working standard measurements was always smaller than half the compatibility goal, indicating that calibrations were made with sufficient frequency. Similar stability was achieved during all measurement campaigns.

The optimal calibration frequency was further investigated with laboratory experiments. For this purpose, a gas standard (dry

natural air) was continuously measured using one of the Picarro G2401 WCC-Empa travelling instruments. To prolong the length of the measurement period, the sample flow was reduced to 30 ml/min by a needle valve at the inlet port, which still allowed the stabilization of the cavity pressure to 140 torr (186.6 hPa). The standard gas was calibrated against NOAA/ESRL standards before and after the experiment to ensure that no drift occurred over the observation period, which



may happen when a standard gas is losing pressure (Leuenberger et al., 2015). The initial pressure of the standard was 129.6 bar and dropped to 62.1 bar at the end of the experiment. Figure 3 summarises the results of these measurements. The upper left panel shows the $CO_2$ variation of the standard gas (5 s raw data) during the experiment duration of 455 hours (approx. 19 days). A slight upward drift was observed, which we consider as instrumental drift since the (secondary) standard has been

proven to be stable with respect to the NOAA standards over the course of the experiment. The variations for methane are shown in the upper right panel; again, instrument drift was observed, but in contrast to $CO_2$, no monotonous trend was detected. The lower panels of Figure 3 show the Allan standard deviation for $CO_2$ and $CH_4$ (Werle et al., 1993) using the data above, which allows an estimate of the optimal calibration intervals. Based on this experiment, the optimal averaging time is approximately 20 min for both $CO_2$ and $CH_4$. This is shorter than the $CO_2$ minimum at 58 min that was found by

Flowers et al. (2012), but compares well with the results of Yver Kwok et al. (2015). However, the Allan standard deviation only slightly increases up to 1e+5 s (27.8 h); therefore, a calibration interval of 30 hours is regarded as an optimal compromise with regard to stability, data coverage and the consumption of calibration gas.

On-site comparison experiments with the travelling instrument were made at the following GAW stations:

(i) Danum Valley, Malaysia (DMV), operated by the Malaysian Meteorological Department, Kuala Lumpur, Malaysia, with

support from Commonwealth Scientific and Industrial Research Organisation, Oceans & Atmosphere, Aspendale, Victoria, Australia; (ii) Pallas, Finland (PAL), operated by the Finnish Meteorological Institute, Helsinki, Finland; (iii) Cape Verde Atmospheric Observatory (CVO), operated by the National Institute of Meteorology and Geophysics, Cabo Verde, with support from the Max Planck Institute for Biogeochemistry, Jena, Germany, and the University of York, United Kingdom; (iv) Mace Head, Ireland (MHD), operated by the National University of Ireland, Galway, Ireland.

All comparison experiments with the WCC-Empa travelling instrument were made using a separate inlet system, i.e. a separate air sampling tubing line (¼" OD Synflex 1300 or ½" OD at DMV) leading to the same air intake location as the station. This WCC-Empa inlet line was flushed by an additional pump at a flow rate of approximately 2 L/min. At PAL, the travelling instrument switched occasionally to the single station inlet. CVO has two separate inlets: reactive gases are sampled 8 m above the ground on top of the measurement container, whereas the GHG inlet is located on top of a tower 30

25   m above the ground. Comparisons with the WCC traveling instrument at CVO were performed by switching occasionally to the inlet for reactive gases. It should be noted that only comparisons were selected where neither the WCC-Empa travelling instrument nor the station analyser had instrumental problems. Furthermore, comparisons between the travelling instrument and other CRDS analysers are not shown in this paper, since the scope of the current work focuses on the comparison of CRDS with those techniques that have been widely used in the past. Results of the WCC-Empa travelling instrument and

another CRDS instrument at PAL were published by Rella et al. (2013). An overview of the comparisons, including duration and instruments, is presented in Table 1. More information on the stations is available from the GAW Station Information System (GAWSIS, 2016).



## 3 Results

The $CO_2$ and $CH_4$ ambient air comparison experiments selected for this study were carried out at four GAW stations (cf. Table 1). The selected sites span a range of climatologies from tropical to subarctic conditions. Furthermore, $CO_2$ and $CH_4$ variability were distinctly different, ranging from remote baseline conditions at CVO with almost no temporal variation to highly variable conditions due to atmosphere – biosphere exchange processes at DMV. Figure 4 shows the diurnal variations for $CO_2$ and $CH_4$ as well as the frequency distribution of $CO_2$, $CH_4$ and $H_2O$ measured by the travelling instrument during the 1-2 month long campaigns. Almost no or little diurnal variation was observed at the remote stations CVO and PAL, whereas the measurements at MHD and DMV showed significantly more variability because of sporadic signals from $CO_2$ and $CH_4$ source regions and due to vegetation uptake and respiration. The selected campaigns also cover different atmospheric water vapour contents, ranging from dry sub-arctic conditions at PAL ($H_2O$ <1%) to temperate (MHD, $H_2O$ 1-2%) and tropical (DMV, $H_2O$ > 2%) conditions. At CVO, the air sampled with the travelling instrument was dried with a Nafion dryer, because carbon monoxide (CO) was also studied during this particular campaign (not shown here), and the water vapour correction for CO at that time and for this specific instrument prevented sufficiently precise humid CO measurements to be made by the travelling instrument (Zellweger et al., 2012).

### 3.1 Carbon Dioxide Ambient Air Comparisons

Figures 5 - 7 show $CO_2$ comparisons made at the PAL, DMV and CVO GAW stations. In the upper panels, the comparison with the highest available common time resolution (1 min) is shown, together with the deviation of the station instrument compared to the travelling instrument as function of time, and a histogram of the observed bias. The middle panels show the same but for hourly aggregated values where all available data were considered. The lower panels also show hourly aggregates, but mean values were calculated using only high resolution data with concurrent data availability of the travelling and station instruments.

The temporal variation was well captured at all stations even at the highest time resolution of one minute. To account for different residence times in the inlet system, data of the travelling instrument was slightly shifted (up to 53 s) to obtain the best possible agreement between the time series. The mean $CO_2$ bias based on 1 min data was 0.08 ± 0.06 ppm at PAL, 0.01 ± 0.67 ppm at DMV, and 0.06 ± 0.08 ppm at CVO. These deviations are in good agreement with the results obtained when WCC-Empa travelling standards were measured on the station analysers as summarised in Figure 8, where the bias is shown for individual travelling standards (black dots), including a linear regression analysis with 95% confidence intervals. The resulting deviation is very close to the bias observed during the ambient air comparison, which is also shown as small red points (hourly data) in Figure 8. Normally, the performance audit covers a wider mole fraction range compared to the ambient variability of a station. This gives valuable information about either the compensation of the instrument non-linearity, the consistency of the used standards or a combination of both, which would not be available from the ambient air comparison alone. The above results indicate that the non-linearity of the analysers was well corrected at PAL and DMV. In





contrast, the travelling standard comparison shows a larger bias at CVO. Most likely, this is due to the fact that the travelling standards are significantly out of the CVO calibration range, which is narrow in response to the small variability of the $CO_2$ mole fraction at this station.

The mean agreement and standard deviation remained almost unchanged after aggregation to hourly values at all stations, indicating that the procedures for time synchronization and sample residence time correction were appropriate. The data availability was different for the travelling instrument and the station instruments mainly due to different requirements concerning calibration frequency of the station analysers. The CRDS travelling instrument was normally calibrated every 30 hours for 45 min, which results in a very high data availability and a uniform data coverage. In contrast, the NDIR $CO_2$ instruments at PAL and DMV require more frequent calibrations, which have been implemented using different approaches. The data availability was more or less homogeneous over time at PAL, whereas at DMV the hourly average $CO_2$ mole fraction value is calculated using only the final 44 minutes of each hour due to the automated hourly zero drift correction mode employed at the start of every hour to monitor short-term detector drifts. For OA-ICOS instrument at CVO, a scheme alternating between a working standard and ambient air measurement in an interval of 1 min was in place resulting in a homogenous data availability of every second minute. This has been done to account for short-term drifts. The data availability of the station analysers and the travelling instrument as a function of the minute of the hour as well as the minute of the day is shown in Figure 9. Data coverage becomes extremely important with rapid atmospheric changes, e.g. during flux measurements (Peltola et al., 2014), and at sites influenced by local biospheric processes.

The influence of the data coverage was assessed by the calculation of hourly averages with concurrent data availability for the highest time resolution, which is shown in the lower panels of Figures 5 - 7. As expected, no influence was found for the CVO data series, which is due to the combination of very small ambient air variability and homogeneous data coverage of the CVO analyser. Some effect was observed at PAL, where the standard deviation of the mean bias changes from 0.05 to 0.03 ppm if only concurrent data points are considered for averaging. It should be noted that in this case all occurrences with a bias of >0.2 ppm disappear. These results can be compared to the simultaneous comparison exercise of the travelling instrument with an onsite CRDS instruments (Rella et al., 2013), where also a mean standard deviation of the difference between the two instruments of 0.03 ppm was observed. As expected, the effect of data coverage on hourly mean values was the most pronounced at DMV due to the largest ambient variability and the omission of the first 16 minutes of data from each hourly calculation. The standard deviation of the mean bias decreased significantly from 0.62 to 0.21 ppm when only concurrent data was considered for the aggregation of hourly means. In order to test the influence of the temporal coverage, we also calculated the difference between hourly averages of the travelling instrument using all available data and only data of the travelling instrument with concurrent data of the DMV instrument. The resulting distribution of the differences looks very similar to the bias observed between the travelling instrument and the DMV analyser, with a standard deviation of 0.54 ppm. Therefore, the differences that we observe between the DMV analyser and the travelling instrument originate almost entirely from different temporal coverage. The bias induced by the data coverage can well exceed the extended WMO/GAW compatibility goals for hourly values in case of rather large variability of the ambient air $CO_2$ mole fraction. As reported





here, homogeneous data coverage over the period of consideration can be more important than the absolute data availability. As an example, measurements made every second minute (CVO) better characterise an hourly average compared to a setup where the first 16 minutes of each hour is not measured (DMV). Consequently, in the case of reduced data coverage due to the applied calibration scheme or analytical technique the resulting uncertainties cannot be neglected but should be evaluated

and reported along with the uncertainty budget of the analysis.

## 3.2   Methane Ambient Air Comparisons

Figures 10 and 11 show $CH_4$ comparison experiments made at the CVO and MHD GAW stations in the same format as Figures 5 to 7 for $CO_2$. At CVO, the $CH_4$ data coverage is the same as for $CO_2$, since it is measured with the same OA-ICOS instrument (Los Gatos Research, LGR- GGA-24EP), whereas the GC/FID system (Carle 100A) at MHD analyses only two

ambient air samples per hour, resulting in a maximal data availability of 3.3 % if single injections are being considered to be representative for one minute. The actual data availability at MHD however was 2.2 % due to further instrument down times. The observed $CH_4$ bias between CVO and WCC-Empa is -0.61±0.49 ppb based on 1 min data, well within the WMO GAW compatibility goal of ±2 ppb. Further averaging to hourly values slightly reduces the standard deviation of the bias to 0.34 ppb, whereas similarly to $CO_2$ no significant improvement is observed if only concurrent high resolution data are considered

for the hourly aggregate. The scatter of the observed bias is significantly larger for the MHD comparison, averaging to -0.57±3.79 ppb based on comparison with single injections of the MHD GC/FID with concurrent travelling instrument 1 min data. This is slightly larger compared to the difference that was observed during a 2-month comparison campaign by Vardag et al. (2014), when a mean bias to a FTIR of -0.04±3.38 ppb was observed. The standard deviation however is comparable, and a significant part of it can be attributed to the repeatability of the GC/FID system at MHD. The observed relative

standard deviation under repeatability conditions (multiple injections of travelling standard) was 0.12% during the audit, which results in a scatter of 2.25 ppb for the travelling instrument-GC/FID comparison. The aggregation of hourly values nearly doubles the scatter to ±4.20 ppb. This can be expected since two single injections per hour lead to poor statistics and do not sufficiently reflect the observed variability of ambient $CH_4$ at MHD.

As for $CO_2$, the results of the comparison of the travelling standards during audit were confirmed by $CH_4$ ambient air

comparisons both at CVO and MHD. The deviations were in good agreement with the results of the travelling standard comparison. This is shown in Figure 12, where the bias is plotted for individual travelling standard (black dots), including a linear regression analysis with 95% confidence intervals. Also here, the bias observed during the ambient air comparison (orange points in Figure 12) agrees well with the bias determined by the performance audit. In case of CVO, the observed scatter during the ambient air measurement was comparable with the 95% confidence bands of the linear interpolation of the

audit results, whereas the ambient air scatter at MHD was larger. Again, this is expected due to the different data coverage of the different techniques.



### 3.3 Influence of the Inlet System

During the comparison experiments at PAL and CVO the travelling instrument switched occasionally from the WCC inlet to the station inlet. This was used to further investigate the influence of the inlet system. At PAL, no difference was observed between the station and the WCC-Empa inlet systems for $CO_2$. The vertical grey shaded areas in Figure 5 denote the periods when the travelling instrument sampled air from the PAL inlet, whereas the dedicated WCC-Empa inlet was used the rest of the time. The difference between the two inlets was not significant (0.01±0.06 ppm), indicating that the PAL inlet system is fully appropriate. This was also the case for the CVO $CH_4$ measurements, where the mean bias of the CVO instrument was -0.71±0.50 ppb measured at the CVO inlet compared to -0.59±0.49 ppb at the WCC-Empa inlet. However, a small $CO_2$ bias was observed at CVO, with a mean deviation of 0.05±0.06 ppm measured at the CVO GHG inlet compared to 0.13±0.09 ppm at the reactive gases inlet. This is not unexpected, since the reactive gases inlet is located 24 m below the CVO GHG inlet, and thus may be more influenced by the local and regional $CO_2$ emissions. Again, this result indicates that the inlet system is fully appropriate at CVO as well.

In other cases (Zellweger et al., 2013), which are not shown here due to the scope of the present study, significant deviations between inlet systems can be observed. Those were mainly caused by leaks or inefficient drying procedures. The use of independent inlet systems during ambient air comparisons can clearly provide valuable information on the overall performance of the measurement set-up, which cannot be obtained by the comparisons of standard gases alone. In the cases presented here, the inlet designs were appropriate and did not contribute to a potential bias.

### 3.4 Water Vapour Correction

Traditional measurement techniques for GHGs require drying of the sample gas to achieve the WMO/GAW compatibility goals. A recent study by Rella et al. (2013) shows that measurements of humid air can be made using CRDS instruments if an appropriate correction for water vapour interference is applied. They concluded that it is possible to make $CO_2$ and $CH_4$ measurements within the GAW compatibility goals (for the Northern Hemisphere) for water vapour levels up to at least 2% by determining the water vapour correction function once on a per-instrument basis. During our study, all WCC-Empa measurements, except at CVO, were made without drying the air sample.

The water vapour correction functions were experimentally determined several times per instrument, which provides information about the stability and reproducibility of the correction function. The correction function is a second order polynomial, as shown by following equations:

$$CO_2(dry) = CO_2(wet) / (1 + a*H_2O + b*H_2O^2) \tag{1}$$

$$CH_4(dry) = CH_4(wet) / (1 + c*H_2O + d*H_2O^2) \tag{2}$$

$CO_2(wet)$, $CH_4(wet)$ and $H_2O$ are the (humid) mole fractions in ppm ($CO_2$, $CH_4$) or % ($H_2O$) reported by the analyser.



Figure 13 shows the differences between the first and the subsequent measurements for the two instruments at a nominal mole fraction of 400 ppm ($CO_2$) and 1900 ppb ($CH_4$) with the WMO/GAW compatibility targets. The CRDS instruments used for this study have built in water vapour correction functions and report also dry mole fractions. However, to achieve the best possible correction equation, it is advisable that instrument specific correction functions are determined for each instrument using the method described by Rella et al. (2013). The robustness of the correction can further be improved by pooling a number of correction functions obtained by several experiments, or by selecting a correction function that is representative for a larger set of experiments.

Figure 13 shows that the $CO_2$ correction functions result in dry mole fractions that are within the WMO/GAW compatibility goals for water vapour up to 2% if functions obtained at the same day are pooled (CFKADS#2001), and up to >2.5% for the newer analyser (CFKADS#2098), which is in line with the results published by Rella et al. (2013). The differences between determinations of the water correction function at same day and after longer time periods were similar, which indicates that the short-term and the long-term sensitivity changes are in the same order, or that the repeatability of the droplet test is the limiting factor. Also, it was confirmed that the $CH_4$ correction functions result in dry mole fractions that are within the WMO/GAW compatibility goals for water vapour up to 3%. In our case, only one correction function that was initially retrieved was used to correct the data. These initial coefficients, as determined using the Empa method described by Rella et al. (2013), are summarised in Table 2.

The difference between dry station instrument measurements and humid travelling instrument measurements as a function of water vapour is illustrated in Figure 14 for all comparisons. No dependency was found between the observed $CO_2$ bias and $H_2O$ measured by the travelling instrument at PAL. This result is consistent with a comparison using the data of the travelling instrument and another CRDS instrument with dry sample air that has been published by Rella et al. (2013). It is noteworthy that the same result was obtained at DMV despite the much higher $H_2O$ content of the ambient air. A small remaining dependency cannot be excluded based on the current study due to the high variability of the observed bias; however, the contribution to the overall uncertainty would be small compared to other sources of uncertainty. Since the water vapour interference is the limiting factor for the $CO_2$ measurements due to the uncertainty caused by the correction, it is important to note that the results of PAL and particularly DMV confirm the applicability of $CO_2$ correction functions.

### 3.5 Evaluation of Audit Results

With our study, we aimed to evaluate the performance of modern spectroscopic analysers (CRDS, OA-ICOS, and FTIR) in comparison to the traditional techniques. WCC-Empa conducted thirty-two station audits with a travelling standard for $CH_4$ (2005-2014), and twenty for $CO_2$ (2010-2015). Each comparison was evaluated by linear regression analysis as shown in Figure 12. To judge whether the resulting slope / intercept combinations meet the WMO/GAW compatibility and extended compatibility goals, the bias in the centre of the mole fraction range (405 ppm for $CO_2$, 1900 ppb for $CH_4$) of the unpolluted troposphere (WMO, 2014) (360-450 ppm for $CO_2$, 1700-2100 ppb for $CH_4$) was plotted against the slope of the individual



travelling standard comparisons. This is shown in Figure 15 along with the allowed bias / slope combinations corresponding to the compatibility (dark grey area) and extended compatibility goals (light grey area) over the entire mole fraction range of the unpolluted troposphere. Only comparisons that were on the same calibration scale and without any known instrument malfunctions were considered. It can be clearly seen that large differences exist among the evaluated analytical techniques.

Newer spectroscopic techniques such as CRDS and OA-ICOS show generally better performance with respect to accuracy and measurement uncertainty compared to NDIR ($CO_2$) and GC/FID ($CH_4$). Moreover, these techniques also provide better data coverage, which further reduces the uncertainty.

The results of the above analysis are further presented in Figure 16, which summarises the percentage of comparisons that met the compatibility and extended compatibility goals. We show all comparisons and then separately only travelling

standard – CRDS, travelling standard – NDIR ($CO_2$) and travelling standard – GC/FID ($CH_4$) comparisons.

It is obvious that reaching the compatibility goals for $CO_2$ remains a challenge; out of the twenty comparisons, only two (10%) met the compatibility goal with seven (35%) meeting the extended goal. However, these results include the entire $CO_2$ mole fraction range relevant for the troposphere. Often, the calibration ranges at stations are intentionally limited to the ambient mole fraction ranges typical for their location. Such ranges can be significantly smaller than those used for Figures

15 and 16, e.g. at CVO. Therefore, slope/bias pairs that are outside the compatibility goals do not necessarily imply that the measurements at a station are biased. However, they provide useful information about the performance of the instrument as well as the calibration over the entire mole fraction range of the unpolluted troposphere.

For $CO_2$, the overall compatibility of the travelling standard – CRDS comparisons was significantly better compared to the travelling standard – NDIR comparisons. From the total nine travelling standard – CRDS comparisons, five (56%) were

20 within ±0.2 ppm, and one within ±0.1 ppm (11%). Out of the eight travelling standard - NDIR comparisons, one (12%) reached the ±0.1 ppm and another one (12%) the ±0.2 ppm limit. All other comparisons (75%) were outside the compatibility goals. This is primarily due to the poorer repeatability and limitations of the NDIR technique in general (drift and non-linearity of the detector), and secondly, the calibration linearity issues which become relevant when comparison results are expanded beyond the typical mole fraction range for a particular measurement site. This was also reflected by the

25 error bars in Figure 15, which in most cases were significantly larger for travelling standard –NDIR comparisons compared to travelling standard – CRDS $CO_2$ comparisons.

Meeting the WMO/GAW compatibility goals for $CH_4$ is significantly less challenging. Out of the thirty-two comparisons, seventeen (53%) were within the compatibility goals, and ten (31%) were within the extended compatibility goals. A total number of five (16%) comparisons did not meet the extended compatibility goals.

Figure 16 presents the results analysed as a function of the measurement techniques. All $CH_4$ travelling standard - CRDS comparisons were within the extended compatibility goals; nine (82%) of the comparison met the ±2 ppb limit, whereas the ±5 ppb limit was met in the remaining two (18%) cases. This is significantly more than for the GC/FID systems, of which only eight (44%) met the ±2 ppb limit, and five (28%) the ±5 ppb limit. The remaining five (28%) of the $CH_4$ travelling



standard - CRDS comparisons were outside the compatibility goals. Focusing on the station relevant $CH_4$ mole fraction range should not be the reason of the worse result compared to CRDS, since the GC/FID method is known to be linear in the mole fraction range discussed here. More likely, worse instrument repeatability compared to CRDS plays an important role here, which is further illustrated by the significantly larger error bars of the bias / slope pairs in Figure 15 for GC/FID systems.

Other techniques (OA-ICOS, FTIR) also indicate better repeatability compared to GC/FID ($CH_4$) and NDIR ($CO_2$); however, an insufficient number of comparison studies have been made to reliably show the superior performance of these techniques.

# 4 Conclusions

The results of the analysis of both travelling standards and side-by-side comparisons with a travelling instrument show that laser-based spectrometers such as the Picarro CRDS can be suitable for accurate $CO_2$ and $CH_4$ measurements. This is important with respect to the continuation of long-term time series of $CO_2$ and $CH_4$. Due to the higher temporal data coverage, repeatability and linearity, the accuracy of $CO_2$ and $CH_4$ measurements is expected to improve when traditional technologies such as NDIR analysers for $CO_2$ and GC/FID systems for $CH_4$ are replaced. Furthermore, CDRS derived $CO_2$ and $CH_4$ measurements do not require the drying of the sampled air and water vapour corrections appear to be valid even under very humid conditions as encountered at tropical sites. The resulting remaining uncertainty is assumed to be small compared to other contributing factors to the overall uncertainty. In particular, incomplete data coverage unavoidable for quasi-continuous methods (e.g. GC measurements) or techniques requiring frequent calibrations (e.g. NDIR measurements), remains one of the significant contributing factors to the overall uncertainty. This becomes most important at locations with high temporal variability of the observed parameters, e.g. due to local and regional emission sources or atmosphere-biosphere exchange processes.

A thorough analysis of the $CO_2$ and $CH_4$ stability of CRDS instruments indicates that the optimal calibration frequency is approximately 30 hours. This frequency is sufficient to compensate for the instrumental drift, and at the same time it allows to reduce the loss of ambient air measurements when calibrating an instrument. We believe that the modern measurement techniques such as CRDS will increase the number of GAW stations complying with the WMO/GAW compatibility goals for both $CO_2$ and $CH_4$. However, the fact remains that the compatibility goal of ±0.1 ppm for $CO_2$ can still be very challenging for many stations. Out of the measurement techniques employed in this study, the CRDS analyser has shown the best performance. Next to this specific type of laser spectrometer thoroughly assessed here, other techniques such as FTIR or OA-ICOS are becoming more widely available. To date, not enough comparison data are available from our audits to draw firm conclusions on the performance of these techniques, but initial results indicate that they also have the potential of being superior compared to traditional methods (NDIR, GC/FID). However, despite the advantages of the new techniques, care has to be taken with regard to calibration strategies, sample inlet set-up and appropriate water vapour corrections.



Our analysis has demonstrated that, providing an adequate design of the measurement system, the performance assessment with either travelling standards or a travelling instrument leads to a similar conclusion. The two comparison methods supply complementary information: the approach which utilises travelling standards is better suited to characterise the performance of the instrument with regard to e.g. linearity, whereas the side-by-side comparison has the advantage of incorporating the

5   whole system including an inlet, a drying system and the instrument calibration over a longer period of time. Therefore, this two-pillar audit scheme with comparison of travelling standards and multi-week side-by-side comparisons of station instrumentation with travelling instruments proves to be a valid approach for data quality assessments at atmospheric measurement stations.

10   **Acknowledgements.** This work was supported by MeteoSwiss through engagement in the Global Atmosphere Watch programme. The authors would like to thank Picarro Inc. for providing a travelling instrument. We further acknowledge the support by the station staff at various GAW stations during the audits, in particular by Luis Mendes and Helder Lopez at CVO. Financial support for the GHG measurements at CVO is provided by the German Max-Planck-Society and by the University of Exeter, UK.





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



Table 1: Overview of $CO_2$ and $CH_4$ comparison experiments presented in this study.

| Location | Coordinates | Start | End | Station instrument | Travelling Instrument | Compound |
|---|---|---|---|---|---|---|
| PAL | 67.973 N 24.116 E | 2012-04-31 | 2012-06-09 | LI-COR LI-7000 | CFKADS2001 humid meas. | $CO_2$ |
| DMV | 4.981 N 117.844 E | 2013-12-06 | 2014-02-25 | LoFlo Mark II | CFKADS2098 humid meas. | $CO_2$ |
| CVO | 16.864 N 24.868 W | 2012-12-12 | 2013-02-04 | LGR GGA-24EP | CFKADS2001 dry meas. | $CO_2$ / $CH_4$ |
| MHD | 53.325 N 9.900 W | 2013-07-24 | 2013-08-27 | CARLE 100A GC/FID | CFKADS2098 humid meas. | $CH_4$ |

Table 2: Water vapour correction coefficients for the WCC-Empa travelling CRDS instruments; coefficients a and b refer to the correction of $CO_2$, while c and d refer to $CH_4$ (see equations 1 and 2).

| Analyser | Date | a | b | c | d |
|---|---|---|---|---|---|
| CFKADS2001 | 2012-01-30 | -0.015331 | 0.000062 | -0.012452 | 0.000065 |
| CFKADS2098 | 2013-06-27 | -0.015625 | 0.000095 | -0.013050 | 0.000175 |



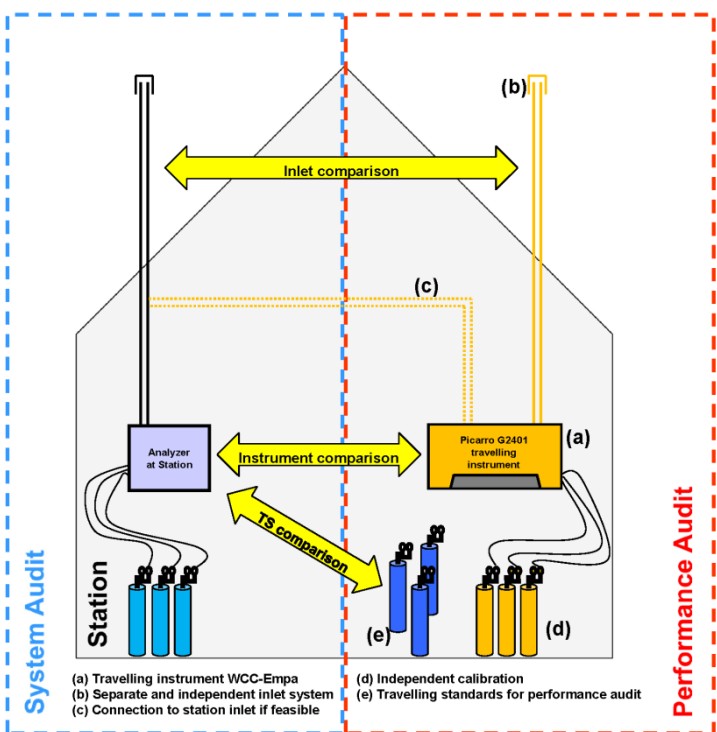

Figure 1. Schematic of the comparison procedure for the ambient air measurements during audits by WCC-Empa.




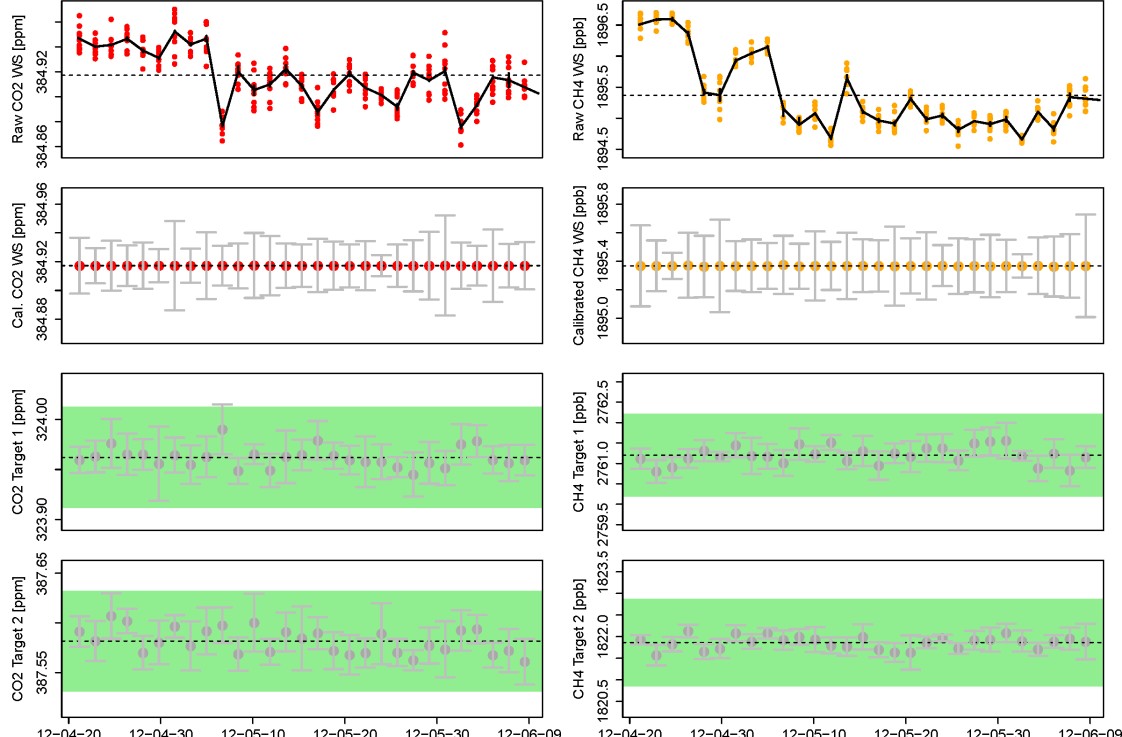

Figure 2. Left-hand side panels: $CO_2$ working and target cylinder measurements of the travelling instrument at PAL. Top panel: Raw 1 min readings of the WS (red points) with LOESS fit (solid black line) and the mean reading (dotted grey line). Second panel from the top: Average WS readings after calibration. Lower two panels: Target cylinder measurements. The green area represents the average reading ± half of the WMO compatibility goal. Right-hand side panels: Same results for $CH_4$.





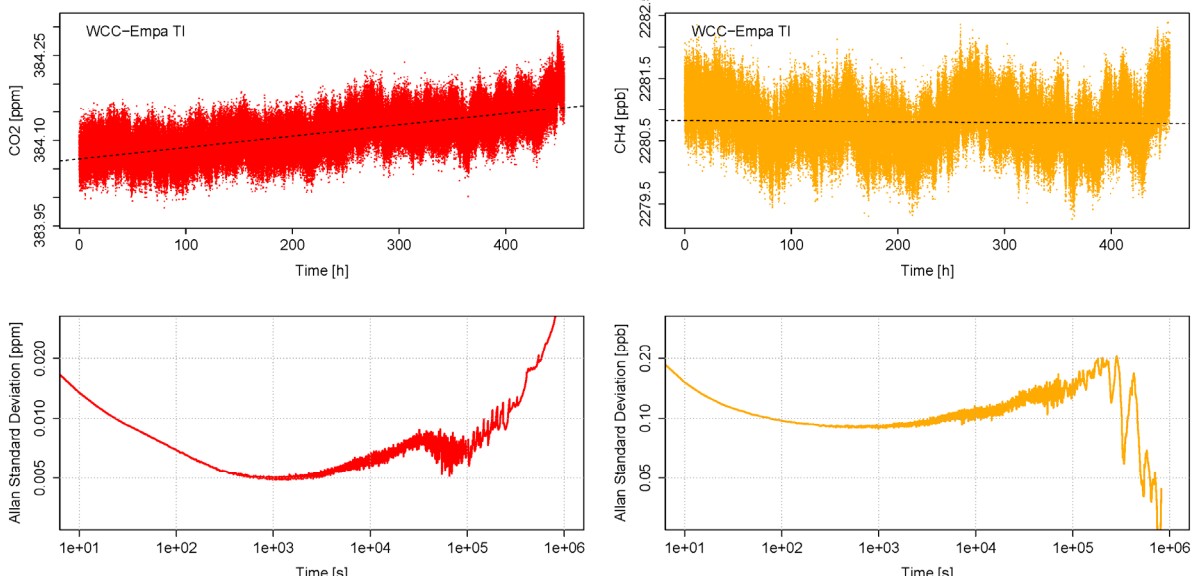

Figure 3. Upper left panel: $CO_2$ working standard measured over a period of 455 hours with the Picarro G2401 travelling instrument (5 second averages). The black dashed line is the linear regression through all data. Lower left panel: Allan standard deviation plots based on the data above. Right: Same for $CH_4$.



Figure 4. (a) Mean $CO_2$ diurnal variation measured during the comparison campaigns at CVO, PAL, MHD and DMV with the travelling instrument. The error bars are the standard deviation of each hourly value. (b) Frequency distribution of hourly $CO_2$ mole fractions (bin size 0.5 ppm). (c) Same as (a) for $CH_4$. (d) Same as (b) for $CH_4$, bin size 1 ppb. (e) Frequency distribution of the hourly $H_2O$ content of the atmosphere (bin size 0.05 %), except for CVO, where the $H_2O$ content after the Nafion dryer is shown.





Figure 5. CO$_2$ comparison at PAL between the WCC-Empa travelling instrument and the PAL LI-COR LI-7000 instruments. Left: CO$_2$ time series and CO$_2$ bias vs time. Right: Deviation histogram. Upper set: 1 min data; Middle set: 1-h data, calculated from all available 1 min values; Lower set: 1-h data, calculated from 1 min values with concurrent PAL and WCC-Empa data. The grey areas correspond to the WMO/GAW compatibility (dark grey) and extended compatibility (light grey) goals; vertical grey bars (left diagrams) illustrate when different inlets were used (see text for details).



Figure 6. $CO_2$ comparison at DMV between the WCC-Empa travelling instrument and the DMV LoFlo Mark II instruments. Left: $CO_2$ time series and $CO_2$ bias vs time. Right: Deviation histogram. Upper set: 1 min data; Middle set: 1-h data, calculated from all available 1 min values; Lower set: 1-h data, calculated from 1 min values with both DMV and WCC-Empa data coverage. The grey areas correspond to the WMO/GAW compatibility (dark grey) and extended compatibility (light grey) goals.





Figure 7. $CO_2$ comparison at CVO between the WCC-Empa travelling instrument and the CVO LGR GGA-24EP instruments. Left: $CO_2$ time series and $CO_2$ bias vs time. Right: Deviation histogram. Upper set: 1 min data; Middle set: 1-h data, calculated from all available 1 min values; Lower set: 1-h data, calculated from 1 min values with concurrent CVO and WCC-Empa data from both inlets. The grey areas correspond to the WMO/GAW compatibility (dark grey) and extended compatibility (light grey) goals; vertical grey bars (left diagrams) illustrate when different inlets were used (see text for details).





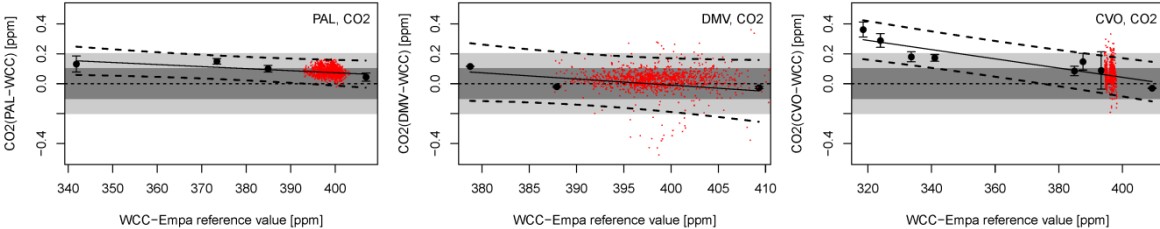

Figure 8. Results of the $CO_2$ travelling standard comparisons (performance audits) at PAL (left), DMV (middle) and CVO (right). The grey areas correspond to the WMO/GAW compatibility (dark grey) and extended compatibility (light grey) goals. The red points correspond to the observed differences based on hourly data during the ambient air comparison. Solid and dashed lines represent the fit and the 95% confidence intervals for the linear regression through the travelling standard comparison.





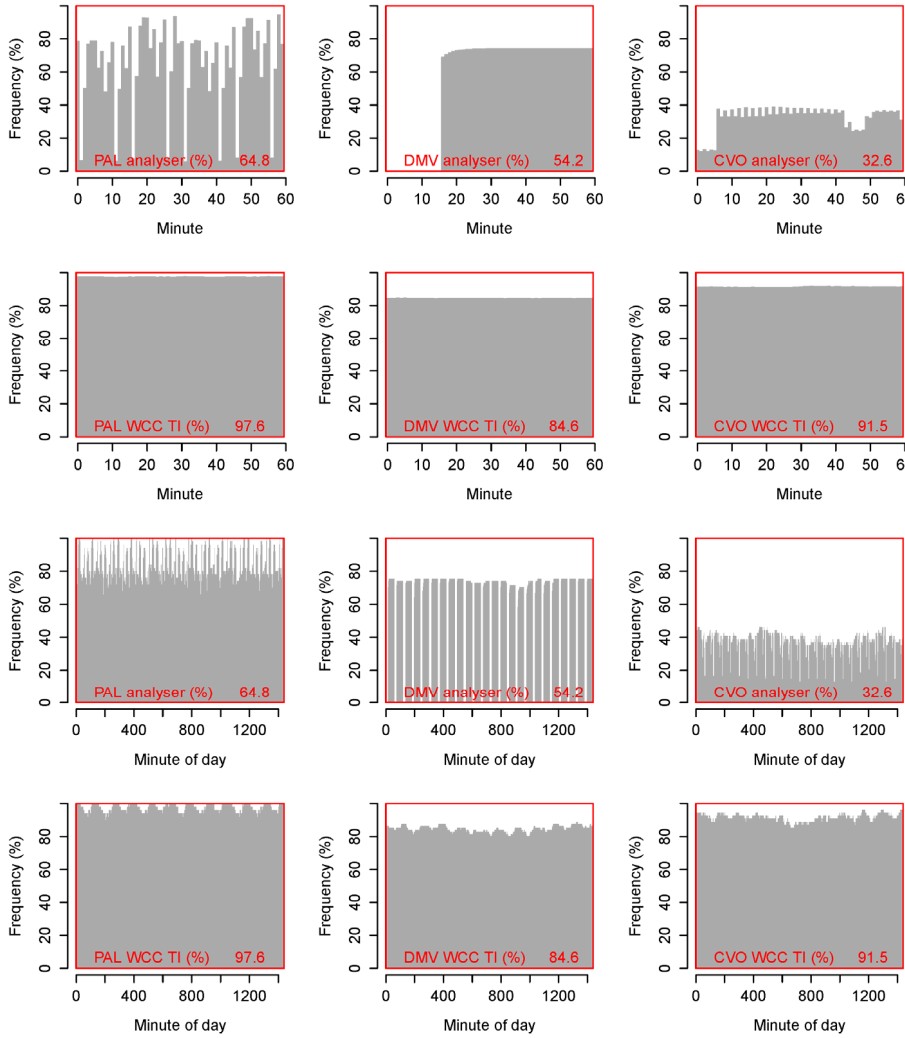

Figure 9. First row: mean $CO_2$ data availability of the PAL, DMV and CVO station instruments as a function of the minute of the hour. Second row: same as above, for the WCC-Empa travelling instrument. Third row: mean $CO_2$ data availability for the station instruments as a function of the minute of the day. Fourth row: same as above, for the WCC-Empa travelling instrument.



Figure 10. $CH_4$ comparison at CVO between the WCC-Empa travelling instrument and the CVO LGR GGA-24EP instruments. Left: $CH_4$ time series and $CH_4$ bias vs time. Right: Deviation histogram. Upper set: 1 min data; Middle set: 1-h data, calculated from all available 1 min values; Lower set: 1-h data, calculated from 1 min values with both CVO and WCC-Empa data coverage. The grey areas correspond to the WMO/GAW compatibility (dark grey) and extended compatibility (light grey) goals; vertical grey bars illustrate when different inlets were used (see text for details).





Figure 11. CH$_4$ comparison at MHD between the WCC-Empa travelling instrument and the MHD CARLE GC/FID instruments. Left: CH$_4$ time series and CH$_4$ bias vs time. Right: Deviation histogram. Upper set: 1 min / single injection data; Middle set: 1-h data, calculated from all available 1 min and single injection values; Lower set: 1-h data, calculated from 1 min and single injection values with both MHD and WCC-Empa data coverage. The grey areas correspond to the WMO/GAW compatibility (dark grey) and extended compatibility (light grey) goals.





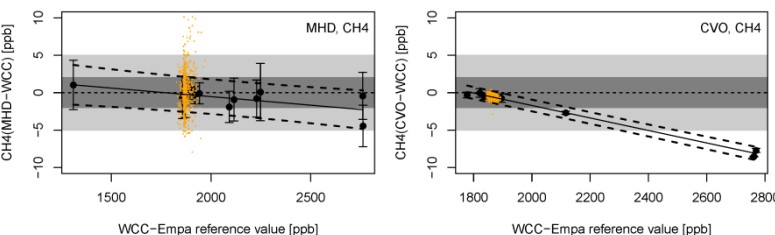

Figure 12. Results of the $CH_4$ travelling standard comparisons (performance audits) at MHD (left) and CVO (right). The grey areas correspond to the WMO/GAW compatibility and extended compatibility goals. The orange points correspond to the observed differences based on hourly data during the ambient air comparison.

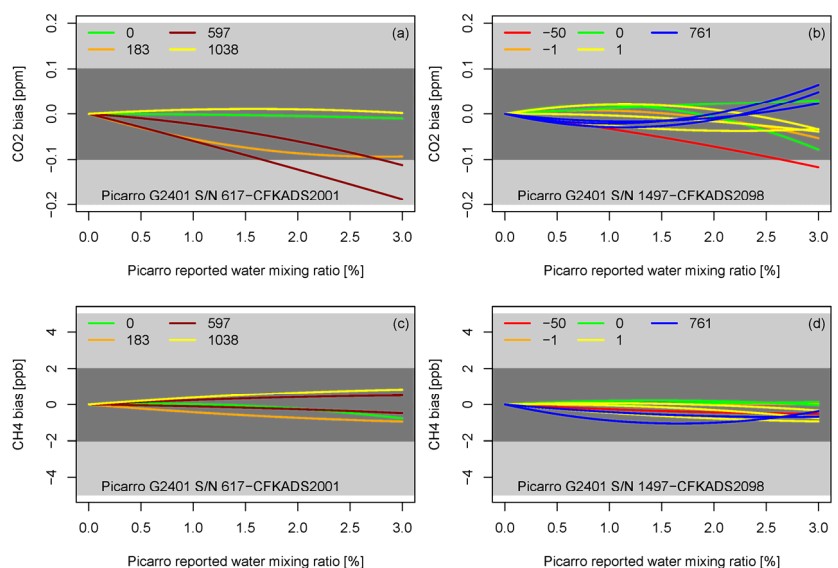

Figure 13. (a) Bias at 400 ppm $CO_2$ compared to initial correction function of the Picarro G2401 S/N 617-CFKADS2001. The different lines correspond to the individual determinations of the correction functions. The number in the legend refers to the number of days from the determination of the reference correction function. (b) Same as (a), for Picarro G2401 S/N 1497-CFKADS2098. (c) and (d): Same as (a) and (b), for $CH_4$ at 1900 ppb. The grey areas correspond to the WMO/GAW compatibility (dark grey) and extended compatibility (light grey) goals.





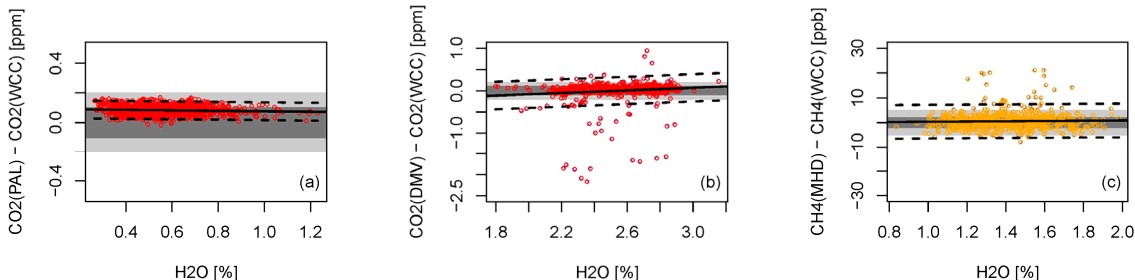

Figure 14. (a) $CO_2$ bias of PAL vs. $H_2O$ measured by the travelling instrument based on hourly matched data. The solid black lines denote the linear regressions, while the dashed lines are the 95% confidence bands. (b) Same as (a), for DMV. (c): Same as (a), for $CH_4$ at MHD. The grey areas correspond to the WMO/GAW compatibility and extended compatibility goals.

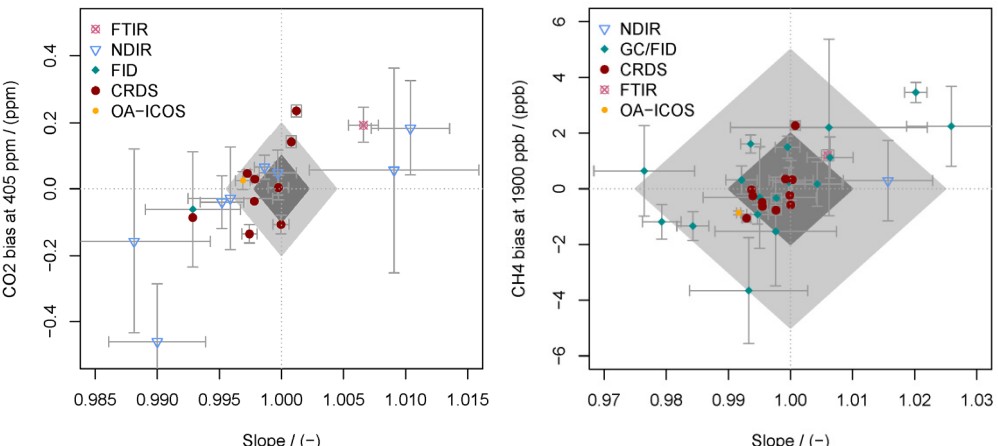

Figure 15. Left: $CO_2$ bias at 405 ppm vs. the slope of the performance audit for individual travelling standard comparisons involving different measurement techniques of the audited station analysers. Right: Same for $CH_4$, bias at 1900 ppb. The dark grey and light grey areas correspond to the WMO/GAW compatibility and extended compatibility goals for the range from 1700 – 2100 ppb $CH_4$ and 360 - 450 ppm $CO_2$.





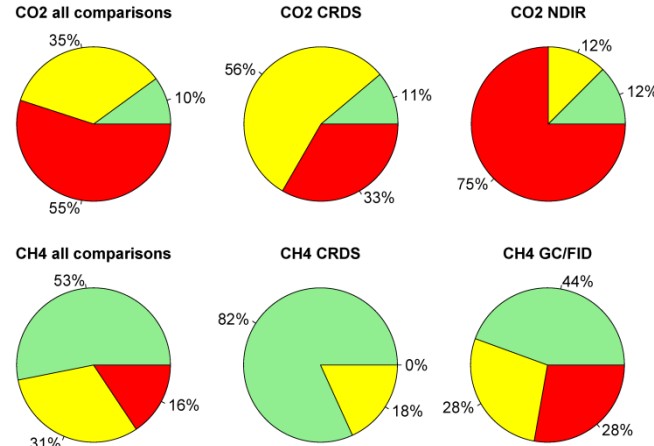

Figure 16. Upper panel: Percentage of $CO_2$ performance audit results that were within the WMO/GAW compatibility goals in the range from 360-450 ppm (green), the extended compatibility goals (yellow), or outside the compatibility goals (red area). Results for all, only CRDS and only NDIR comparisons are shown. Lower panel: Same for $CH_4$ in the range of 1700-2100 ppb, for all, only CRDS and only GC/FID comparisons.