# Peer review of "Assessment of recent advances in measurement techniques for atmospheric carbon dioxide and methane observations"

_Atmospheric Measurement Techniques, 2016_

## Referee Comment (RC1) · Anonymous Referee #1 · 30 May 2016

General comments: This study presents a comparison between historically used methods to measure CO2 and CH4 (NDIR and GC mostly) and new techniques such as CRDS, OA-ICOS or FTIR with a focus on Picarro CRDS. They show that the new techniques lead to improved accuracy with reduced bias and representativity of the data. The paper focuses on four sites where the comparisons took place but also uses data from comparing tank measurements that are more numerous. The results are very interesting to many users that will have to transition from "older" to "newer" techniques. The referee recommends publication in AMT.

I only have two questions: Page 4 l9: Could you briely detail the Empa method? At some point later, there is a mention of a droplet test, is that what you do? Page 11 l11

to17: Did you recalculate the slope of the comparison when restraining your dataset to the cylinders with concentrations within the station calibration set range? Does that change the results towards the compatibility goals? It seems to me that station PI would appreciate to know if within the range they meet the compatibility goals and not just that they don't on a broader range.

---

## Referee Comment (RC2) · Anonymous Referee #2 · 19 Jun 2016

This paper compares historic, established techniques for measuring atmospheric methane and carbon dioxide (GC/FID and NDIR) with a suite of spectroscopic techniques (in particular cavity ringdown spectroscopy (CRDS)), rapidly being taken up by researchers measuring greenhouse gases. The work assesses the suitability of the new methods for delivering a seamless continuation of the CO2 & CH4 timeseries within the WMO/GAW network. Through a series of extensive 'audits' (intercomparisons between instruments) at four sites, along with a set of measurements of circulating standards, the authors conclude that spectroscopic techniques are not only suitable for replacing traditional methods, but in fact offer a considerable improvement over traditional methods, improving accuracy and reducing bias.

[Figure]

This paper describes some and interesting and highly relevant work for the atmospheric greenhouse gas measurement community and the reviewer recommends publication int AMT.

---

## Editor Comment (EC1) · D.W.T. Griffith (Editor) · 1 Jul 2016

Both reviewers recommend publication with only minor techncal corrections. I encourage the authors to adddress these corrections in a revised manuscript. In addition I have a couple of editor comments, general and technical:

Editor general comment:

I am confused by the references to the "audit" in this paper. It seems that there are two distinct studies here, and they are intermixed in a way that is (to me) unclear. Firstly there is the travelling instrument side-by-side measurements of ambient air at 4 stations, which is the main thrust of the paper, occupies most of the experimental

description and results up to section 3.5. Secondly section 3.5 then describes a quite different study for which there is no corresponding description in section 2 - Experimental. This is the "audit", which appears to be a separate study in which travelling standards were circulated around a much larger suite of stations and measurement methods over a longer time period. This is quite a separate study and requires some introduction to the sites and instruments and methods in section 2. (I am aware independently that this study happened, but not the details.) At present there seems to be only the last sentence of section 1, and the first two of 3.5, to introduce the "audit". Without further explanation, Figure 15 is very difficult to interpret. Firstly, there is no description of the sites and instruments included in this comparison. Secondly, in a 2-parameter linear regression the slope and intercept can trade off against each other, so the value of plotting the only the slope against the "bias" is not at all clear. Some further explanation is required.

My suggestion is therefore to modify the paper to separate the descriptions and results of the two comparisons . Unfortunately this will make the paper longer, and the authors might reconsider if the audit study could be removed from this paper and published separately; in its current form it is incomplete. If expanding the current paper to address these concerns , I suggest: 1. Provide a separate subsection of 2. Experimental to describe the audit and distinguish the two studies. All details of the audit required to interpret the results to be presented should be included. 2. Arrange the results sections in part 3 to relate to section 2 so that the results of the travelling instrument and the audit are separated. Figures 8, 12 and 15 relate to the audit, not the travelling instrument, and should be grouped together with the relevant description of results. Parallel and contrasting conclusions from both studies can then be made.

Editor technical corrections

The manuscript is well produced and I have only two technical comments:

- Page 5 line 6 replace "monotonous" with "monotonic".

- P5 line 7 and many subsequent examples. I think the correct term should "Allan deviation", not "Allan standard deviation" This is not a "standard deviation" in the statistical sense, and should not be confused with the usual standard deviation. It is the square root of the the Allan Variance, which is something quite different from the usual statistically-defined variance. Perhaps there is a formal definition of this nomenclature somewhere, but I am not aware of it.
* * *

---

## Author Comment (AC1) · 15 Jul 2016

**Reply to referees and the editor for the manuscript amt-2016-110 "Assessment of recent advances in measurement techniques for atmospheric carbon dioxide and methane observations" by Christoph Zellweger et al.**

**Referee #1**

We would like to thank Referee #1 for the valuable comments and her/his time to review the manuscript. Our replies are below.

**I only have two questions: Page 4 l9: Could you briefly detail the Empa method? At some point later, there is a mention of a droplet test, is that what you do?**

Yes. The Empa method has already been described in detail in Zellweger et al. (2012), and we added this reference at the relevant position in the revised manuscript (Page 4, Line 10). We added the following brief description:

*Experimental details of the Empa method are described in Zellweger et al. (2012). Briefly, a small amount of water (approximately 0.8 ml) was directly injected into a constant flow (approximately 500 ml min−1) of a working standard which was delivered to the instrument. The resulting water vapour influence was then fitted by a quadratic function.*

We also clarified that the droplet test was made according to the Empa method (Page 10, Line 13), and cite the corresponding references (Rella et al., 2013, and Zellweger et al., 2012).

**Page 11 l11 to17: Did you recalculate the slope of the comparison when restraining your dataset to the cylinders with concentrations within the station calibration set range? Does that change the results towards the compatibility goals?**

We did not recalculate the slope using restrained mole fraction ranges. This would certainly change the results towards the compatibility goals for some of the comparisons. However, since the scope of our paper was the assessment of different measurement techniques under well-defined conditions, we chose the same mole fraction range for all comparisons.

**It seems to me that station PI would appreciate to know if within the range they meet the compatibility goals and not just that they don't on a broader range.**

Yes, this is absolutely true. The results of our comparisons are always communicated in a way that considers the mole fraction range relevant for a measurement station. For this study we widened the range in order to have a common and well-defined comparison criterion for the assessment of the different techniques.

**Referee #2**

We also would like to thank Referee #2 for her/his time to review the manuscript. Referee #2 didn't have specific comments that need to be addressed here.

**Editor comments**

We would like to thank Dave Griffith his time to carefully read the manuscript and for his valuable comments.

**I am confused by the references to the "audit" in this paper. It seems that there are two distinct studies here, and they are intermixed in a way that is (to me) unclear. Firstly there is the travelling instrument side-by-side measurement of ambient air at 4 stations, which is the main thrust of the paper, occupies most of the experimental description and results up to section 3.5. Secondly section 3.5 then describes a quite different study for which there is no corresponding description in section 2 - Experimental. This is the "audit", which appears to be a separate study in which travelling standards were circulated around a much larger suite of stations and measurement methods over a longer time period. This is quite a separate study and requires some introduction to the sites and instruments and methods in section 2. (I am aware independently that this study happened, but not the details.) At present there seems to be only the last sentence of section 1, and the first two of 3.5, to introduce the "audit".**

We would like to thank Dave Griffith for this comment. We agree that the distinction between the two different approaches for the performance audit was not clear enough. We addressed this by adding a more detailed description of the two methods in the experimental section, with two subchapters on the 'Performance audit using travelling standards' and on 'Performance audit by parallel measurements with a travelling instrument'.

While we left the description of the performance audit by parallel measurements mostly unchanged (now new section '2.2 Performance audit by parallel measurements with a travelling instrument', starting with line 24 / page of the original manuscript, we added more details to the introduction of the experimental section and also to the new section '2.1 Performance audit using travelling standards'.

The new parts (in italic) are as follows:

2. Experimental

The quality assurance strategy of the GAW programme comprises system and performance audits (hereafter only called audit) carried out by World Calibration Centres (WCCs). *WCC-Empa is the designated WCC for $CH_4$ (since 2000) and $CO_2$ (since 2010) audits. The performance audits conducted by WCC-Empa are made using two different approaches. The first method, which is described in Section 2.1 below, is based on the comparison of travelling standards (calibrated standard gases). This method has been an integral part of all performance audits made by WCC-Empa since we started this activity in 1995. In addition to the comparisons of travelling standards, a second approach by parallel measurements using a travelling instrument was implemented more recently. The latter approach, which is described in more detail ins Section 2.2, was introduced after it was recognised that standard comparisons alone often lack important sources of potential biases, for example effects in the air inlet system.*

*2.1 Performance audit using travelling standards*

The concept of the audit procedure *using travelling standards* has been described in detail elsewhere (Buchmann et al., 2009; Klausen et al., 2003). In brief, an audit involves the comparison of travelling standards

(i.e. compressed gas in high pressure cylinders) on the analytical system of the audited station (WMO, 2011b). The travelling standards are calibrated against primary laboratory reference standards traceable to the Central Calibration Laboratory (CCL) before and after the audit. The audited station's personnel analyse the travelling standards and report the mole fractions which are compared to the values assigned by the WCC. The result is analysed by a linear regression between the reference (WCC) and the station values. For the calibration of the travelling standards at WCC-Empa, a GC/FID (Varian 3800) system was used from 2000 to 2009 for $CH_4$; from 2009 a CRDS (Picarro Inc., G1301 $CO_2/CH_4/H_2O$ analyser) has been used for both $CH_4$ and $CO_2$ calibrations. Several standards of the CCL (NOAA/ESRL, National Oceanic and Atmospheric Administration / Earth System Research Laboratory) are used as reference standards at WCC-Empa ensuring traceability to the CCL.

*For the current study we analysed performance audit results for methane (2005-2014) and carbon dioxide (2010-2015). Details of the comparisons including instruments and analytical techniques are given in Table 2 for $CO_2$ and Table 1 for $CH_4$. In order to assess the performance of the individual comparisons in a standardised way, the bias in the centre of the mole fraction range (405 ppm for $CO_2$, 1900 ppb for $CH_4$) of the unpolluted troposphere (WMO, 2014) (360-450 ppm for $CO_2$, 1700-2100 ppb for $CH_4$) was calculated for these comparisons based on the linear regression analysis. This allows displaying the result of a performance audit using travelling standards as a single dot in a bias vs. slope plot, as illustrated in Figure 1 for the example of $CO_2$ audits. The green dashed line in the left panel of Figure 1 shows a case with no bias at 405 ppm $CO_2$ but with the corresponding minimal slope that is possible for the data still meeting the data quality objective (DQO) of 0.1 ppm in the range of 360-450 ppm $CO_2$. This case translates to a single point in the bias vs. slope plot, as shown by the green dot in the left panel of Figure 1. For illustrative purpose, two additional cases are shown: the maximum allowed bias with the corresponding slope that still meets the extended DQO of 0.2 ppm (orange dashed line / dot), and a case with a slope / bias combination that does not meet the DQOs (red dashed line / dot) over the entire relevant mole fraction range.*

**Without further explanation, Figure 15 is very difficult to interpret. Firstly, there is no description of the sites and instruments included in this comparison.**

We agree that further details are required for the understanding of Figure 15. We added the following two tables that give details of the sites and instruments included in the comparisons.

Table 1: CO$_2$ performance audits using travelling standards from 2010 to 2015

| Station | GAW ID | Year | Instrument | Method | Intercept (ppm) | Slope (-) | Bias at 405 ppm CO$_2$ (ppm) |
|---|---|---|---|---|---|---|---|
| Lauder | LAU | 2010 | FTIR | FTIR | -2.48 | 1.00660 | 0.19 |
| Cape Point | CPT | 2011 | Hartmann & Braun URAS 4 | NDIR | 4.65 | 0.98813 | -0.16 |
| Zugspitze | ZSF | 2011 | HP6890 | GC/FID | 2.83 | 0.99286 | -0.06 |
| Hohenpeissenberg | HPB | 2011 | Picarro G1301 | CRDS | -0.09 | 0.99996 | -0.11 |
| Bukit Koto Tabang | BKT | 2011 | Picarro G1301 | CRDS | 2.81 | 0.99285 | -0.09 |
| Pallas | PAL | 2012 | Picarro G2401 | CRDS | 0.85 | 0.99781 | -0.04 |
| Pallas | PAL | 2012 | LI-COR LI-7000 | NDIR | 0.62 | 0.99863 | 0.07 |
| Zeppelin Mountain | ZEP | 2012 | Picarro G2401 | CRDS | -0.25 | 1.00120 | 0.24 |
| Zeppelin Mountain | ZEP | 2012 | LI-COR LI-7000 | NDIR | 3.59 | 0.99000 | -0.46 |
| Cape Verde | CVO | 2012 | LGR GGA-24EP | OA-ICOS | 1.28 | 0.99690 | 0.02 |
| Cape Verde | CVO | 2012 | Siemens Ultramat 6F | NDIR | 0.17 | 0.99970 | 0.05 |
| Mace Head | MHD | 2013 | Picarro G1301 | CRDS | 0.90 | 0.99785 | 0.03 |
| Mace Head | MHD | 2013 | Picarro G2301 | CRDS | 1.16 | 0.99725 | 0.05 |
| Izaña | IZO | 2013 | LICOR LI-7000 | NDIR | 1.90 | 0.99521 | -0.04 |
| Izaña | IZO | 2013 | LICOR LI-6252 | NDIR | -4.02 | 1.01038 | 0.18 |
| Danum Valley | DMV | 2013 | LoFlo Mark II | NDIR | 1.64 | 0.99588 | -0.03 |
| Bukit Koto Tabang | BKT | 2014 | Picarro G1301 | CRDS | 0.91 | 0.99742 | -0.13 |
| Anmyeon-do | AMY | 2014 | Picarro G2301 | CRDS | -0.18 | 1.00079 | 0.14 |
| Jungfraujoch | JFJ | 2015 | Picarro G2401 | CRDS | 0.10 | 0.99975 | 0.00 |
| Jungfraujoch | JFJ | 2015 | SICK MAIHAK S710 | NDIR | -3.62 | 1.00907 | 0.06 |

Table 2: CH$_4$ performance audits using travelling standards from 2005 to 2014

| Station / Laboratory | GAW ID | Year | Instrument | Method | Intercept (ppb) | Slope (-) | Bias at 1900 ppb CH$_4$ (ppb) |
|---|---|---|---|---|---|---|---|
| Ryori | RYO | 2005 | Horiba GA-360 | NDIR | -29.6 | 1.0157 | 0.29 |
| Japan Meteorological Agency | NA | 2005 | Shimadzu 14BPF | GC/FID | 2.4 | 0.9995 | 1.49 |
| Zugspitze | ZSF | 2006 | HP 6890 | GC/FID | 9.1 | 0.9933 | -3.66 |
| Jungfraujoch | JFJ | 2006 | Aglient 6890 | GC/FID | -9.6 | 1.0062 | 2.20 |
| Cape Point | CPT | 2006 | Varian CP-3800 | GC/FID | -47.0 | 1.0259 | 2.25 |
| Pallas | PAL | 2007 | Agilent 6890N | GC/FID | 15.3 | 0.9921 | 0.32 |
| Barrow | BRW | 2008 | HP 6890 | GC/FID | 38.2 | 0.9793 | -1.18 |
| Izaña | IZO | 2009 | DANI-3800 | GC/FID | 9.1 | 0.9950 | -0.31 |
| Mt. Waliguan | WLG | 2009 | HP 5890 | GC/FID | 3.0 | 0.9976 | -1.53 |
| Mt. Waliguan | WLG | 2009 | Aglient 6890 | GC/FID | 0.1 | 1.0001 | 0.31 |
| Mt. Waliguan | WLG | 2009 | Picarro G1301 | CRDS | 3.7 | 0.9977 | -0.77 |
| GAW calibration lab Beijing | NA | 2009 | Agilent 6890N | GC/FID | 28.5 | 0.9843 | -1.34 |
| GAW calibration lab Beijing | NA | 2009 | Agilent 6890N | GC/FID | 13.8 | 0.9936 | 1.61 |
| GAW calibration lab Beijing | NA | 2009 | Picarro G1301 | CRDS | 0.8 | 1.0008 | 2.27 |
| Mace Head | MHD | 2009 | CARLE 100A | GC/FID | 0.7 | 0.9998 | 0.24 |
| Lauder | LAU | 2010 | FTIR | FTIR | -10.2 | 1.0060 | 1.20 |
| Cape Point | CPT | 2011 | Varian CP-3800 | GC/FID | -34.9 | 1.0202 | 3.46 |
| Zugspitze | ZSF | 2011 | HP6890 | GC/FID | 9.2 | 0.9947 | -0.92 |
| Hohenpeissenberg | HPB | 2011 | Picarro G1301 | CRDS | -0.2 | 1.0003 | 0.33 |
| Bukit Koto Tabang | BKT | 2011 | Picarro G1301 | CRDS | -0.6 | 1.0000 | -0.57 |
| Pallas | PAL | 2012 | Picarro G2401 | CRDS | 12.4 | 0.9929 | -1.05 |
| Zeppelin Mountain | ZEP | 2012 | Picarro G2401 | CRDS | 11.3 | 0.9939 | -0.25 |
| Mt. Cimone | CMN | 2012 | Agilent 6890N | GC/FID | 45.4 | 0.9764 | 0.64 |
| Cape Verde | CVO | 2012 | LGR GGA-24EP | OA-ICOS | 15.0 | 0.9917 | -0.86 |
| Mace Head | MHD | 2013 | CARLE 100A | GC/FID | 4.0 | 0.9977 | -0.33 |
| Mace Head | MHD | 2013 | Picarro G1301 | CRDS | 8.2 | 0.9954 | -0.48 |
| Mace Head | MHD | 2013 | Picarro G2301 | CRDS | 11.9 | 0.9937 | -0.03 |
| Izaña | IZO | 2013 | DANI 3800 | GC/FID | -11.0 | 1.0064 | 1.12 |
| Izaña | IZO | 2013 | Varian 3800 | GC/FID | -8.0 | 1.0043 | 0.17 |
| Bukit Koto Tabang | BKT | 2014 | Picarro G1301 | CRDS | 0.0 | 0.9999 | -0.23 |
| Anmyeon-do | AMY | 2014 | Picarro G2301 | CRDS | 8.0 | 0.9955 | -0.63 |
| Jungfraujoch | JFJ | 2015 | Picarro G2401 | CRDS | 1.9 | 0.9992 | 0.36 |

**Secondly, in a 2-parameter linear regression the slope and intercept can trade off against each other, so the value of plotting only the slope against the "bias" is not at all clear. Some further explanation is required.**

We added a new figure that explains the concept in more detail, as well as some explanatory text (see new parts in experimental section above). It is correct that slope and bias can trade off against each other. However, the slope / bias combinations that are allowed to meet the compatibility goals within a given mole fraction range is well defined.

[Figure]

*Figure 1. Left: Deviation vs. reference value plot for CO$_2$ (illustrative) for three different cases (green, orange, red; details see text) for the mole fraction of 360 - 450 ppm CO$_2$. Right: Illustrative bias vs. slope plot for the cases shown in the left panel (details see text). The grey areas correspond to the WMO/GAW compatibility (dark grey) and extended compatibility (light grey) goals.*

**My suggestion is therefore to modify the paper to separate the descriptions and results of the two comparisons .**

We addressed this with the above modifications.

**Unfortunately this will make the paper longer, and the authors might reconsider if the audit study could be removed from this paper and published separately; in its current form it is incomplete. If expanding the current paper to address these concerns , I suggest: 1. Provide a separate subsection of 2. Experimental to describe the audit and distinguish the two studies.**

Done, see above.

**All details of the audit required to interpret the results to be presented should be included.**

Done, see above.

**2. Arrange the results sections in part 3 to relate to section 2 so that the results of the travelling instrument and the audit are separated. Figures 8, 12 and 15 relate to the audit, not the travelling instrument, and should be grouped together with the relevant description of results. Parallel and contrasting conclusions from both studies can then be made.**

We would like to keep the current order of the figure. It is true that Figures 8 and 12 also relate to the performance audit using travelling standards, but we also show how this compares with the ambient air comparison in the same figure. The value of showing this together in one figure is to demonstrate that the two different audit approaches lead to the same result if the whole measurement set-up is appropriate.

**Editor technical corrections The manuscript is well produced and I have only two technical comments:**

**Page 5 line 6 replace "monotonous" with "monotonic".**

Done.

**P5 line 7 and many subsequent examples. I think the correct term should "Allan deviation", not "Allan standard deviation" This is not a "standard deviation" in the statistical sense, and should not be confused with the usual standard deviation. It is the square root of the Allan Variance, which is something quite different from the usual statistically-defined variance. Perhaps there is a formal definition of this nomenclature somewhere, but I am not aware of it.**

We changed to Allan Deviation, as suggested.

**Additional comment received by e-mail:**

One reader brought to our attention that recently another manuscript comparing a Picarro CRDS (G2301 Model) against a GC-FID for CH4 measurements was published (http://pubs.acs.org/doi/abs/10.1021/ac5043076). This work is complementary to ours by reinforcing its metrology aspect. We therefore included the following reference in the revised version of our manuscript on Page 2, Line 25:

Flores, E., Rhoderick, G. C., Viallon, J., Moussay, P., Choteau, T., Gameson, L., Guenther, F. R., and Wielgosz, R. I.: Methane Standards Made in Whole and Synthetic Air Compared by Cavity Ring Down Spectroscopy and Gas Chromatography with Flame Ionization Detection for Atmospheric Monitoring Applications, Analytical Chemistry, 87, 3272-3279, 2015.

**Other corrections:**

The GAW Station Information System migrated to a new server. We updated the corresponding reference / hyperlink.